# Modified Benzoxazole-Based VEGFR-2 Inhibitors and Apoptosis Inducers: Design, Synthesis, and Anti-Proliferative Evaluation

**DOI:** 10.3390/molecules27155047

**Published:** 2022-08-08

**Authors:** Alaa Elwan, Abdallah E. Abdallah, Hazem A. Mahdy, Mohammed A. Dahab, Mohammed S. Taghour, Eslam B. Elkaeed, Ahmed B. M. Mehany, Ahmed Nabeeh, Mohammed Adel, Aisha A. Alsfouk, Hazem Elkady, Ibrahim H. Eissa

**Affiliations:** 1Pharmaceutical Medicinal Chemistry & Drug Design Department, Faculty of Pharmacy (Boys), Al-Azhar University, Cairo 11884, Egypt; 2Department of Pharmaceutical Sciences, College of Pharmacy, AlMaarefa University, Riyadh 13713, Saudi Arabia; 3Zoology Department, Faculty of Science (Boys), Al-Azhar University, Cairo 11884, Egypt; 4Department of Pharmaceutical Sciences, College of Pharmacy, Princess Nourah bint Abdulrahman University, Riyadh 11671, Saudi Arabia

**Keywords:** anticancer, benzoxazole, molecular modeling, VEGFR-2 kinase

## Abstract

This work is one of our efforts to discover potent anticancer agents. We modified the most promising derivative of our previous work concerned with the development of VEGFR-2 inhibitor candidates. Thirteen new compounds based on benzoxazole moiety were synthesized and evaluated against three human cancer cell lines, namely, breast cancer (MCF-7), colorectal carcinoma (HCT116), and hepatocellular carcinoma (HepG2). The synthesized compounds were also evaluated against VEGFR-2 kinase activity. The biological testing fallouts showed that compound **8d** was more potent than standard sorafenib. Such compound showed IC_50_ values of 3.43, 2.79, and 2.43 µM against the aforementioned cancer cell lines, respectively, compared to IC_50_ values of 4.21, 5.30, and 3.40 µM reported for sorafenib. Compound **8d** also was found to exert exceptional VEGFR-2 inhibition activity with an IC_50_ value of 0.0554 μM compared to sorafenib (0.0782 μM). In addition, compound **8h** revealed excellent cytotoxic effects with IC_50_ values of 3.53, 2.94, and 2.76 µM against experienced cell lines, respectively. Furthermore, compounds **8a** and **8e** were found to inhibit VEGFR-2 kinase activity with IC_50_ values of 0.0579 and 0.0741 μM, exceeding that of sorafenib. Compound **8d** showed a significant apoptotic effect and arrested the HepG2 cells at the pre-G1 phase. In addition, it exerted a significant inhibition for TNF-α (90.54%) and of IL-6 (92.19%) compared to dexamethasone (93.15%). The molecular docking studies showed that the binding pattern of the new compounds to VEGFR-2 kinase was similar to that of sorafenib.

## 1. Introduction 

Cancer is a life-threatening disease that has been identified as a significant global cause of death, it caused nearly 10 million deaths in the year of 2020 [1]. Many accredited reports have confirmed that lung cancer, colorectal cancer, liver cancer, and breast cancer are the leading causes of cancer death [2]. Globally, cancer’s incidence and death have rapidly increased in the last decades [3]. Therefore, enormous efforts have been made to combat cancer [4,5,6,7,8]. These facts clearly highlight the need for new anticancer drugs that are more effective.

Nonselective chemotherapeutic agents are known to cause severe side effects [9]. Meanwhile, cancer cells differ from normal cells in their specific biochemical abnormalities [10]. The anticancer agents that have been designed to treat such abnormalities are more likely to be potent and more selective. 

A tyrosine kinase abnormality is one of the distinguishing features of cancer cells [11,12]. It was notified that cancer cells have relatively increased levels of vascular endothelial growth factor receptor-2 (VEGFR-2) which has a fundamental role in cancer cell growth [13,14,15]. This is accomplished by supplying cancer cells with new blood carrying oxygen and nutrients (angiogenesis) [16,17,18]. Furthermore, overexpression of VEGFR-2 played an imperative role in the metastasis of solid tumors [19,20]. VEGFR-2 levels were found to be rather high in breast [21], colorectal [22], hepatocellular [23], non-small cell lung [24], and urothelialcarcinomas [25]. Accordingly, anticancer drugs that inhibit VEGFR-2 such as sorafenib [26], regorafenib [27], pazopanib [28], sunitinib [29], tivozanib [30], and lenvatinib [31], are selective and effective against many cancer types [32,33,34,35,36], (Figure 1). 

The FDA has approved many VEGFR-2 inhibitors for use in the clinical management of cancer. However, many complications were reported for such drugs [37,38,39]. In detail, sorafenibexpressed sidee effects such as cardiotoxicity, thrombotic effects, diarrhea, renal impairment and arterial hypertension [40,41]. In the meantime, hepatotoxicity, hand-foot syndrome, GIT disturbance, fatigue, hypertension, decreased appetite, nausea, diarrhea, thrombocytopenia, proteinuria, and hyperparathyroidism were the most common adverse effects of sunitinib, vatalinib, and tivazonib [42,43,44,45].

For many years, our team designed and synthesized many promising VEGFR-2 inhibitors. In 2020, compound **VII** (quinazolin-4(3*H*)-one derivative) was discovered exerting promising activities against three different cancerous cell lines with good VEGFR-2 inhibitory activities. In addition, it induced malignant cell’s apoptosis and arrested their growth at the G2/M phase. In vivo studies revealed that compound **VII** showed a significant tumor growth inhibition effect [46]. In 2021, Some thieno [2,3-*d*]pyrimidine derivatives were synthesized. Compounds **VIII** exhibited the highest cytotoxic activities against HCT-116 and HepG2, with high activity against VEGFR-2 [47]. In the same year, a new series of 3-methylquinoxalin-2(1*H*)-one derived compounds were synthesized. Compounds **IX** [18], and **X** [48], showed the most promising VEGFR-2 inhibitory effects. In addition, these compounds showed good apoptotic induction levels through the elevation of caspase-3, caspase-9, and BAX, in addition to the decrease of Bcl-2 levels. In 2022, compound **XI**, a benzoxazole derivative, decreased the proliferation and the migration potential of HUVEC cells with good VEGFR-2 inhibitory activity. Also, it inhibited the HepG2 cell’s growth at the Pre-G1 phase and induced a significant apoptotic effect [49]. In the same year, compound **XII**, a nicotinamide derivative, expressed the highest anti-proliferative activities against HCT-116 and HepG2, with a potent VEGFR-2 inhibitory effect. The immunomodulatory effect of such compound was assessed against TNF-α and IL-6. It showed a significant decrease in TNF-α level [50], (Figure 2).

This work is an extension of our previous work to discover new VEGFR-2 inhibitors, especially that focused on the benzoxazole-based compounds carrying pharmacophoric structural features of VEGFR-2 inhibitors [49]. We aimed to explore the potency of more related derivatives to establish a more accurate structure–activity relationship, hoping to develop more potent candidates.

### Rationale of the Design

The new derivatives were mainly based on the four reported features of Type II VEGFR-2 inhibitors [48,51,52]. The features comprise a head that fixes the ATP binding domain; spacer; H-bond donor (HBD)/H-bond acceptor (HBA) that binds to the essential Glu885 and Asp1046 amino acid residues at the DFG domain; and hydrophobic moiety occupying the hydrophobic allosteric site that appears just close to the ATP binding domain [53]. It was reported that type II kinase inhibitors are characterized by stabilizing the inactive conformation (DFG-out) of the enzyme by binding to the hydrophobic allosteric site [54]. Therefore, the strong interactions at the allosteric site may increase the stability of the receptor–ligand complex. Consequently, we made further modifications to the allosteric site-interacting terminal phenyl group of the benzoxazole derivatives examined in our previous work [49]. The previously obtained results revealed that the 4-hydroxyphenyl derivative presented in Figure 3 was the most potent candidate against the VEGFR-2 receptor, showing comparable results to sorafenib. Accordingly, it was selected to be a lead compound for the current design (Figure 3). It can be seen that electron-donating (methyl) and electron-withdrawing (nitro and acetyl) groups were designed at the 4-position of the terminal phenyl ring. The three substituents also differ in the hydrophobicity and steric effect. Furthermore, benzyl and phenethyl groups were constructed instead of phenyl to see to what extent the allosteric pocket could interact firmly. In a recent related study, the benzyl group was proved to accommodate the allosteric pocket, showing better results [55].

The anti-cancer potential of the obtained compounds was examined against three human cancer cell lines. The most active candidates were evaluated against VEGFR-2 kinase activity. Then, the most promising member was selected for deep mechanistic studies including apoptosis and cell cycle analyses. In addition, the effects of the most active compounds on the levels of TNF-α and IL-6 were assessed. To reach a good insight into the binding mode of the synthesized compounds against VEGFR-2. Also, a molecular docking study was carried out.

## 2. Results and Discussion

### 2.1. Chemistry

Figure 1, Figure 2 and Figure 3 illustrate the synthetic procedure of the target compounds **8a**–**m**. at first, the 2-aminophenol derivatives **1a**–**c** were cyclized to give the key derivatives in this work (2-mercapto-benzoxazoles **2a**–**c**). The cyclization reactions were performed through the reflux of the mixtures of 2-aminophenols, carbon disulfide, and potassium hydroxide in methanol [56]. To facilitate the nucleophilic substitution reaction, the potassium salts of 2-mercapto-benzoxazoles **3a**–**c** were prepared. These salts were formed through the reflux of compounds **2a**–**c** with alcoholic KOH solution (Figure 1).

In parallel, other key intermediates **7a**–**e** were synthesized in high purity and yields. The sequence of the preparation of compounds **7a**–**e** started with the acylation reaction of 4-aminobenzoic acid **4** with chloroacetyl chloride to give the corresponding acyl form **5**. Then, the carboxylic group of compound **5** was activated via the formation of acyl chloride derivative **6**. This reaction was performed using thionyl chloride and a catalytic amount of DMF as reported [57]. Next, compound **6** was subjected to amide formation (compounds **7a**–**e)** through its stirring with deferent amines in the presence of TEA in acetonitrile as a solvent at room temperature (Figure 2).

The two key groups of intermediates **3a**–**c** and **7a**–**e** were allowed to react to each other in dry DMF over a water bath to produce the final target compounds **8a**–**m** as described in Figure 3.

^1^H NMR charts of the final compounds showed a characteristic peak at about 4.5 ppm for the methylene group that is flanked between S and C = O. The δ value of the spacer amide NH was around 10.8 ppm. The other amide group that was flanked between two phenyl rings showed different δ values according to the substitution at the terminal phenyl ring. The NH attached to the 4-methylphenyl (compounds **8a**–**c**) appeared at about 10 ppm. The NH that attached to the 4-acetylphenyl (compounds **8g**,**h**) appeared at about 10.5 ppm. While the NH attached to the 4-nitrophenyl (compounds **8d**–**f**) was highly deshielded to appear at about 10.8 ppm. The proton of the NH group appeared from 8.5 to 9 ppm when the terminal phenyl was replaced by alkyl phenyl (compounds **8i**–**m**). A peak of three protons at about 2.4 ppm was specific to derivatives based on methybenzoxazole nucleus (compounds **8c**, **f**,**h**, **k**, and **m**). ^13^C NMR charts of all compounds revealed three peaks from about 164 to 167 ppm, for two amides C = O and one C = N of benzoxazole moiety. Compounds **8g**,**h** showed characteristic peaks of ketone C = O at about 197 ppm. IR charts demonstrated signals for NH, amide C = O, aliphatic, and aromatic protons.

### 2.2. Biological Testing

#### 2.2.1. In Vitro Anticancer Activity

The anti-proliferative properties of the synthesized candidates were evaluated against three human cancer cell lines: breast cancer (MCF-7), colorectal carcinoma (HCT116), and hepatocellular carcinoma (HepG2). Sorafenib was utilized as a positive control. The data outlined in Table 1 shows the significance of the new derivatives as anti-proliferative agents. In general, compounds **8d**, **8h**, **8f**, **8e**, **8c**, **8i**, **8k**, and **8a** were the most promising candidates and revealed IC_50_ values ranging from 2.43 to 10.44 µM. In detail, compound **8d** came first, with IC_50_ values of 3.43, 2.79, and 2.43 µM against the mentioned cell lines, respectively. The second active candidate was compound **8h**, which showed IC_50_ values of 3.53, 2.94, and 2.76 µM, respectively. These two anti-proliferative candidates were more potent than sorafenib, which showed IC_50_ values of 4.21, 5.30, and 3.40 µM, respectively. Compound **8f** later showed IC_50_ values of 4.97, 4.00, and 4.41 µM, respectively, which were nearly comparable to those of sorafenib. It can also be noticed that the results of compound **8e** were better than those of sorafenib against HCT116 cells and close to it against MCF-7 and HepG2 cells.

#### 2.2.2. In Vitro VEGFR-2 Kinase Inhibitory Assay

The most promising anti-proliferative candidates: **8d**, **8h**, **8f**, **8e**, **8c**, **8i**, **8k**, and **8a** were further evaluated against VEGFR-2 kinase. The obtained IC_50_ values were listed in Table 2. The most curious result obtained is that three out of the tested eight members were more potent than the reference drug, sorafenib. It can be noticed that compounds **8d** and **8a** showed far better results than sorafenib. They revealed IC_50_ values of 0.0554 and 0.0579 μM, respectively. Furthermore, compound **8e** showed an IC_50_ value of 0.0741 compared to that of sorafenib (0.0782 μM). The other candidates demonstrated IC_50_ values ranging from 0.1142 to 1.139 μM. It is noticeable that compound **8d** was the most potent in both the anti-proliferative assay and the VEGF-2 inhibition assay compared to other candidates and sorafenib as well.

#### 2.2.3. Structure-Activity Relationship

Based on the given results, structure–activity relationships can be established as follows: the candidates based on unsubstituted benzoxazole (**8a**, **8d**, **8i**, and **8l**) were more potent than those based on 5-methyl (**8c**, **8f**, **8h**, **8k**, and **8m**) or 5-chlorobenzoxazole (**8b**, **8e**, **8g**, and **8j**). Also, 5-methylbenzoxazole derivatives were more significant than 5-chlorobenzoxazole derivatives. With respect to the terminal hydrophobic moiety, the strong electron-withdrawing groups were far better than the electron-donating groups. It was found that 4-nitrophenyl derivative (**8d**) showed the best results. Moreover, the 4-acetylphenyl derivative was better than the 4-methylphenyl ones. This indicated that the electron-withdrawing groups as NO_2_ and COCH_3_ are more beneficial for activity than the electron donating one as CH3. This may be due to a decreased partial negative charge on the nitrogen of the amide moiety. This may increase the acidity of the NH group and consequently lead to increase its ability to form a hydrogen bond donor with the carbonyl moiety of Asp1046. In other words, the nitro group can cause the same electronic effect that may be done by the two electron-withdrawing groups (chloro and trifluoromethyl) of sorafenib. Furthermore, compound **8d** (incorporating a nitro group at para position of the terminal phenyl ring) has comparable activity with sorafenib. Substituted phenyl groups were superior to both the benzyl and phenylethyl groups (Figure 4).

#### 2.2.4. HepG2 Cell Cycle Analysis

The most promising candidate, compound **8d**, was further evaluated by analyzing its effect on cycle phases of the HepG2 cells. From the results presented in Table 3, it can be noticed that a large percentage (70.23%) of cells treated with **8d** was accumulated at the pre-G1 phase compared to the 2.70% reported for control cells. This clearly indicates the ability of compound **8d** to inhibit HepG2 cell proliferation efficiently at the pre-G1 phase (Figure 5).

#### 2.2.5. The Effect of 8d on Apoptosis and Necrosis Rates of HepG2

Table 4 and Figure 6 show that the death rate of HepG2 cells treated with compound **8d** was dramatically increased from 2.70% to 70.23%. It was found that the apoptosis rate was sharply increased from 0.60% to 65.22% and from 1.50% to 2.85% at the late and early stages, respectively. Meanwhile, necrosis was increased from 0.60% to 2.16%. This suggests that apoptosis was the main mechanism of HepG2 cell death that was caused by compound **8d**.

#### 2.2.6. The Inhibitory Effect of **8d** on the Cytokines’ Level (TNF-α and IL-6)

Tumor necrosis factor-*α* (TNF-*α*) and interleukin-6 (IL-6) have been shown to have pro-apoptotic properties. The inhibition of these mediators caused apoptosis of tumor cells [58,59]. Therefore, further investigation was conducted to see how compound **8** affect these cytokines (TNF-*α* and IL-6). Compound **8d** was added to HepG2 cells at a concentration of 2.43 µM for 24 h, then the levels of TNF-*α* and IL-6 were measured using the qRT-PCR method. The results showed a significant inhibition of TNF-*α* (90.54%) compared to the control, dexamethasone, which showed an 82.47% inhibition. The data also showed a marked inhibition of IL-6 (92.19%) compared to the control (93.15%). This indicates that compound **8d** has a significant apoptosis induction potential in HepG2 cancer cells through immune-dependent pathways (Table 5).

### 2.3. Molecular Docking

A molecular docking study was performed to recognize the binding interactions by which the designed members bound to their target enzyme [60,61,62,63]. VEGFR-2 kinase (PDB ID: 4ASD) was utilized as a putative binding site in this study. The co-crystallized ligand with the selected protein was sorafenib. Molecular operating environment (MOE 2014) software was used to carry out the docking protocol. Redocking of sorafenib showed an RMSD of 0.24 Å. This value indicatesthe validaty of the employed protocol. Figure 7 declears the overlay of both co-crystallized and redocked sorafenib. The free energies (ΔG) of the binding modes were reported in Table 6.

The binding pattern of the redocked sorafenib is shown in Figure 8. It revealed two hydrogen bonds with Glu885, one hydrogen bond with Asp1046, and one hydrogen bond with Cys919. Additionally, sorafenib interacted via numerous hydrophobic interactions with the different hydrophobic pockets in the active site formed by Leu1035, Ala866 (in the hinge region), Val899, Val916, Lys868, Cys1045 (in the linker region), Leu1019, Leu889, Cys1024, and Ile892 (in the allosteric hydrophobic region).

The obtained results showed that almost all the new candidates fitted the active pocket of VEGFR-2, revealing binding modes similar to that of sorafenib. Figure 9 shows the binding pattern of the most promising candidate **8d**. It can be noticed that the new derivative fitted the active site and took the right orientation through which it formed a hydrogen bond with the essential amino acid Asp1046. Two arene-H interactions were noticed between the benzoxazole moiety and Leu840 at the ATP binding region. In the same region there was also an essential hydrogen bond between the sulfur atom and Cys919. It can be seen that the 4-nitrophenyl group was oriented to occupy the hydrophobic allosteric site of the receptor. In addition, the nitro group at the para position can withdraw the electron from the amide moiety. This may lead to a decrease in the partial negative charge on the nitrogen of the amide moiety and increase the acidity of the NH group. Therefore, there was a good chance of a good fitting in the active site through the formation of a hydrogen bond donor with the carbonyl moiety of Asp1044.

The superimposition of compound **8d** and sorafenib is illustrated in Figure 10, which reveals the similarity of the binding modes of both compound **8d** and sorafenib**.** From Figure 8, it can be noticed that the benzoxazole moiety of compound **8d** has the same orientation of picolinamide moiety of sorafenib. The central phenyl rings of compound **8d** and sorafenib have the same orientation. In addition, the amide group of **8d** has the same position of urea moiety of sorafenib. Furthermore, the hydrophobic tail of compound **8d** (4-nitrophenyl) shows the same orientation of 1-chloro-2-(trifluoromethyl)benzene of sorafenib. This may explain the good biological activity of compound **8d**.

Compound **8a**, the second promising candidate, showed a binding mode similar to that of sorafenib. It revealed the correct orientation of hydrogen bonds with the three essential amino acids Asp1046, Glu885, and Cys919, as can be seen in Figure 11. It also displayed an arene proton interaction between the aromatic ring and NH_2_ of the side chain of Asp923.

Compound **8e** also displayed a similar binding mode as illustrated in Figure 12. It showed hydrogen bonds with the three essential amino acid residues, Asp1046, Glu885, and Cys919. An arene-H interaction was observed between the benzoxazole nucleus and Leu840. It is clear that the new derivative revealed an orientation that was in line with the design. The benzoxazole nucleus was oriented to occupy the ATP binding region. Meanwhile, the para nitrophenyl was directed towards the hydrophobic residues (Ile888, Leu889, Ile892, and Leu1019) in the allosteric region.

## 3. Conclusions

Thirteen benzoxazole-based derivatives were designed and synthesized for anticancer and VEGFR-2 kinase inhibition evaluation. Compound **8d** emerged as the most promising candidate and showed far better results than sorafenib in both anti-proliferative and VEGFR-2 inhibition assays. It showed IC_50_ values of 3.43, 2.79, 2.43, and 0.0554 µM against MCF-7, HCT116, HepG2 cell lines, and VEGFR-2 kinase, respectively. On the other hand, IC_50_ values of sorafenib were 4.21, 5.30, 3.40, and 0.0782 μM, respectively. Furthermore, compounds **8e** and **8a** showed better inhibition of VEGFR-2 kinase than sorafenib. SAR study revealed that unsubstituted benzoxazoles (**8a**, **8d**, **8i**, and **8l**) were more potent than those based on 5-methyl (**8c**, **8f**, **8h**, **8k**, and **8m**) or 5-chlorobenzoxazole (**8b**, **8e**, **8g**, and **8j**). In addition, the substitution of the electron-withdrawing groups with the terminal hydrophobic moiety was beneficial for activity. Furthermore, compound **8d** showed powerful apoptotic effect as well it arrested the HepG2 cell at pre-G1 phase. Moreover, compound **8d** exhibited a significant inhibition for TNF-*α* (90.54%) and of IL-6 (92.19%) compared to the dexamethasone (93.15%). We suggest that these potent candidates, especially **8d**, can serve as a lead compound for further development and discovery of potent anticancer.

## 4. Materials and Methods

### 4.1. Chemistry

#### 4.1.1. General

All apparatus used in the synthesis and analyses were illustrated in the Appendix A. Compounds **3a**–**c** and **7a**–**e** were previously synthesized and reported [49,64,65].

#### 4.1.2. The Preparation of Compounds **8a**–**m**:

A mixture of potassium salts **3a**–**c**, intermediates **7a**–**e**, and KI (0.001 mol of each) in dry DMF (10 mL) was heated for 6 h over a water bath. After reaction’s completion, the mixture was poured over iced water whilst being continuously stirred. Then, the precipitates were filtred and washed with water. The obtained compounds were purified by crystallization from methanol to afford compounds **8a**–**m**. Table 7 shows the colors, yields, and meting points of the obtained compounds.

##### 4-(2-(Benzo[d]oxazol-2-ylthio)acetamido)-N-(p-tolyl)benzamide (**8a**)

IR: 3300 (NH), 3037 (aromatic CH), 3915, 2856 (aliphatic CH), 1660, 1643 (CO amide); ^1^H NMR: 2.27 (s, 3H, CH_3_), 4.47 (s, 2H, CH_2_), 7.15 (m, 2H, Ar-H), 7.33 (m, 2H, Ar-H), 7.68 (m, 6H, Ar-H), 7.99 (m, 2H, Ar-H), 10.11 (s, 1H, NH), 10.78 (s, 1H, NH); Mass (*m/z*): 417 (M^+^), 311, 257, 237, 192, 164, 146, 132, 120 (100%, base peak), 106, 91, 77 and 64.

##### 4-(2-((5-Chlorobenzo[d]oxazol-2-yl)thio)acetamido)-N-(p-tolyl)benzamide (**8b**)

IR: 3426, 3289 (NH), 3064, 3033 (aromatic CH), 2988, 2921, 2857 (aliphatic CH), 1660 (CO amide); ^1^H NMR: 2.28 (s, 3H, CH_3_), 4.48 (s, 2H, CH_2_), 7.14 (d, *J* = 8.2 *Hz*, 2H, Ar-H), 7.37 (dd, *J* = 8.6, 1.7 *Hz*, 1H, Ar-H), 7.67 (m, 6H, Ar-H), 7.96 (d, *J* = 8.2 *Hz*, 2H, Ar-H), 10.09 (s, 1H, NH), 10.76 (s, 1H, NH); ^13^C NMR: 20.97, 37.43, 112.00, 118.52, 118.84, 120.84, 124.78, 129.18, 129.44, 129.47, 130.25, 132.93, 137.18, 141.97, 142.96, 150.62, 165.07, 165.73, 166.42; Mass (*m/z*): 453 (M^+^ + 2), 451 (M^+^), 345, 271, 257, 226, 192, 156, 146, 132, 120 (100 %, base peak), 106, 91, 77 and 63.

##### 4-(2-((5-Methylbenzo[d]oxazol-2-yl)thio)acetamido)-N-(p-tolyl)benzamide (**8c**)

IR: 3428, 3297 (NH), 3030 (aromatic CH), 2985, 2920, 2858 (aliphatic CH), 1662 (CO amide); ^1^H NMR: 2.28 (s, 3H, CH_3_), 2.40 (s, 3H, CH_3_), 4.44 (s, 2H, CH_2_), 7.15 (m, 3H, Ar-H), 7.43 (s, 1H, Ar-H), 7.52 (d, *J* = 8.2 *Hz*, 1H, Ar-H), 7.65 (d, *J* = 8.2 *Hz*, 2H, Ar-H), 7.73 (d, *J* = 8.5 *Hz*, 2H, Ar-H), 7.95 (d, *J* = 8.5 *Hz*, 2H, Ar-H), 10.09 (s, 1H, NH), 10.75 (s, 1H, NH); ^13^C NMR: 20.97, 21.41, 37.29, 110.10, 118.67, 118.83, 120.83, 125.65, 129.17, 129.45, 130.21, 132.93, 134.58, 137.19, 141.88, 142.02, 150.00, 164.18, 165.08, 165.96; Mass (*m/z*): 431 (M^+^), 325, 251, 206, 178, 164, 146, 136, 120 (100 %, base peak), 106, 91, 77 and 65.

##### 4-(2-(Benzo[d]oxazol-2-ylthio)acetamido)-N-(4-nitrophenyl)benzamide (**8d**)

IR: 3414, 3301 (NH), 2921 (aliphatic CH), 1660 (CO amide); ^1^H NMR: 4.46 (s, 2H, CH_2_), 7.33 (m, 2H, Ar-H), 7.64 (m, 2H, Ar-H), 7.77 (d, *J* = 7.4 *Hz*, 2H, Ar-H), 8.01 (m, 4H, Ar-H), 8.24 (d, *J* = 7.9 *Hz*, 2H, Ar-H), 10.71 (s, 1H, NH), 10.81 (s, 1H, NH); ^13^C NMR: 37.34, 110.71, 118.72, 118.90, 120.20, 124.83, 125.15, 125.23, 129.27, 129.63, 141.67, 142.67, 142.77, 146.11, 151.82, 164.31, 165.95, 166.05; Mass (*m/z*): 448 (M^+^), 311, 237, 192, 164, 146, 132, 120 (100%, base peak), 104, 91, 77 and 63.

##### 4-(2-((5-Chlorobenzo[d]oxazol-2-yl)thio)acetamido)-N-(4-nitrophenyl) benzamide (**8e**)

IR: 3290 (NH), 3070 (aromatic CH), 2936 (aliphatic CH), 1687, 1655 (CO amide); ^1^H NMR: 4.48 (s, 2H, CH_2_), 7.36 (d, *J* = 8.1 *Hz*, 1H, Ar-H), 7.71 (m, 4H, Ar-H), 8.01 (m, 4H, Ar-H), 8.25 (d, *J* = 8.5 *Hz*, 2H, Ar-H), 10.72 (s, 1H, NH), 10.82 (s, 1H, NH); ^13^C NMR: 37.44, 112.00, 118.51, 118.90, 120.21, 124.78, 125.26, 129.28, 129.47, 129.64, 142.62, 142.78, 142.95, 146.11, 150.62, 165.85, 165.94, 166.40; Mass (*m/z*): 484 (M^+^ + 2), 482 (M^+^), 345, 271, 226, 198, 179, 156, 146, 132 120 (100%, base peak), 104, 91, 76 and 63.

##### 4-(2-((5-Methylbenzo[d]oxazol-2-yl)thio)acetamido)-N-(4-nitrophenyl) benzamide (**8f**)

IR: 3427, 3290 (NH), 3065 (aromatic CH), 2920, 2853 (aliphatic CH), 1651 (CO amide); ^1^H NMR: 2.52 (s, 3H, CH_3_), 4.44 (s, 2H, CH_2_), 7.12 (d, *J* = 8.0 *Hz*, 1H, Ar-H), 7.42 (s, 1H, Ar-H), 7.51 (d, *J* = 8.0 *Hz*, 1H, Ar-H), 7.77 (d, *J* = 8.0 *Hz*, 2H, Ar-H), 8.02 (m, 4H, Ar-H), 8.26 (d, *J* = 8.5 *Hz*, 2H, Ar-H), 10.73 (s, 1H, NH), 10.81 (s, 1H, NH); ^13^C NMR: 21.40, 37.30, 110.10, 118.66, 118.89, 120.22, 125.27, 125.65, 129.25, 129.64, 134.58, 141.87, 142.68, 142.79, 146.12, 150.08, 164.17, 165.96, 166.08; Mass (*m/z*): 462 (M^+^), 325, 251, 206, 175, 164, 146, 136, 120 (100%, base peak), 104, 91, 78 and 64.

##### N-(4-Acetylphenyl)-4-(2-((5-chlorobenzo[d]oxazol-2-yl)thio)acetamido) benzamide (**8g**)

IR: 3338 (NH), 3078 (aromatic CH), 2924 (aliphatic CH), 1676 (CO ketone), 1654 (CO amide); ^1^H NMR: 2.52 (s, 3H, CH_3_), 4.47 (s, 2H, CH_2_), 7.33 (3, 2H, Ar-H), 7.64 (m, 2H, Ar-H), 7.78 (d, *J* = 7.9 *Hz*, 2H, Ar-H), 8.00 (m, 5H, Ar-H), 10.49 (s, 1H, NH), 10.83 (s, 1H, NH); ^13^C NMR: 26.92, 37.35, 110.70, 118.72, 118.89, 119.86, 124.81, 125.14, 129.48, 129.67, 129.76, 132.33, 141.69, 142.44, 144.23, 151.83, 164.33, 165.70, 166.02, 197.05; Mass (*m/z*): 481 (M^+^ + 2), 479 (M^+^), 345, 226, 192, 179, 156, 146, 132, 120 (100%, base peak), 104, 91, 77 and 63.

##### N-(4-Acetylphenyl)-4-(2-((5-methylbenzo[d]oxazol-2-yl)thio)acetamido) benzamide (**8h**)

IR: 3438, 3306 (NH), 2920, 2853 (aliphatic CH), 1676 (CO ketone), 1653 (CO amide); ^1^H NMR: 2.39 (s, 3H, CH_3_), 2.56 (s, 3H, CH_3_), 4.45 (s, 2H, CH_2_), 7.12 (d, *J* = 8.1 *Hz*, 1H, Ar-H), 7.42 (s, 1H, Ar-H), 7.51 (d, *J* = 8.2 *Hz*, 1H, Ar-H), 7.76 (d, *J* = 8.4 *Hz*, 2H, Ar-H), 7.97 (m, 6H, Ar-H), 10.48 (s, 1H, NH), 10.81 (s, 1H, NH); ^13^C NMR: 21.40, 26.94, 37.31, 110.09, 118.66, 118.88, 119.85, 125.64, 129.47, 129.66, 129.77, 132.35, 134.57, 141.88, 142.44, 144.21, 150.09, 164.17, 165.70, 166.05, 197.06; Mass (*m/z*): 459 (M^+^), 325, 251, 206, 178, 164, 132, 120 (100%, base peak), 104, 91, 78 and 63.

##### 4-(2-(Benzo[d]oxazol-2-ylthio)acetamido)-N-benzylbenzamide (**8i**)

IR: 3438, 3309, 3269 (NH), 3055, 3032 (aromatic CH), 2988, 2927 (aliphatic CH), 1663, 1629 (CO amide); ^1^H NMR: 4.45 (s, 2H, CH_2_), 4.49 (d, *J* = 5.6 *Hz*, 2H, CH_2_), 7.24 (m, 1H, Ar-H), 7.32 (m, 6H, Ar-H), 7.64 (m, 2H, Ar-H), 7.70 (d, *J* = 8.3 *Hz*, 2H, Ar-H), 7.90 (d, *J* = 8.3 *Hz*, 2H, Ar-H), 8.99 (t, *J* = 5.6 *Hz*, 1H, NH), 10.72 (s, 1H, NH); ^13^C NMR: 37.28, 43.05, 110.72, 118.73, 118.87, 124.84, 125.16, 127.18, 127.69, 128.74, 128.77, 129.72, 140.26, 141.68, 141.80, 151.82, 164.33, 165.88, 166.10; Mass (*m/z*): 417 (M^+^), 266, 257, 237, 192, 161, 146, 132, 120, 106, 91 (100%, base peak), 77, 65, 63.

##### N-Benzyl-4-(2-((5-chlorobenzo[d]oxazol-2-yl)thio)acetamido)benzamide (**8j**)

IR: 3431, 3271 (NH), 3033 (aromatic CH), 2923 (aliphatic CH), 1662, 1630 (CO amide); ^1^H NMR: 4.45 (s, 2H, CH_2_), 4.47 (d, *J* = 5.8 *Hz*, 2H, CH_2_), 7.24 (m, 1H, Ar-H), 7.32 (m, 4H, Ar-H), 7.36 (dd, *J* = 8.7 & 1.8 *Hz*, 1H, Ar-H), 7.69 (m, 3H, Ar-H), 7.75 (d, *J* = 1.8 *Hz*, 1H, Ar-H), 7.89 (d, *J* = 8.5 *Hz*, 2H, Ar-H), 8.99 (t, *J* = 5.8 *Hz*, 1H, NH), 10.73 (s, 1H, NH); ^13^C NMR: 37.39, 43.04, 112.01, 118.52, 118.86, 124.78, 127.18, 127.69, 128.74, 128.76, 129.47, 129.73, 140.26, 141.74, 142.97, 150.62, 165.67, 166.07, 166.42; Mass (*m/z*): 453 (M^+^ + 2), 451 (M^+^), 266, 146, 133, 120, 106, 91 (100%, base peak), 77, 65 and 63.

##### N-Benzyl-4-(2-((5-methylbenzo[d]oxazol-2-yl)thio)acetamido)benzamide (**8k**)

IR: 3271 (NH), 3029 (aromatic CH), 2986, 2923, 2869 (aliphatic CH), 1663, 1630 (CO amide); ^1^H NMR: 2.97 (s, 3H, CH_3_), 4.44 (s, 2H, CH_2_), 4.49 (d, *J* = 5.1 *Hz*, 2H, CH_2_), 7.21 (d, *J* = 8.0 *Hz*, 1H, Ar-H), 7.24 (m, 1H, Ar-H), 7.32 (m, 4H, Ar-H), 7.42 (s, 1H, Ar-H), 7.50 (d, *J* = 8.1 *Hz*, 1H, Ar-H), 7.70 (d, *J* = 8.1 *Hz*, 2H, Ar-H), 7.91 (d, *J* = 8.1 *Hz*, 2H, Ar-H), 8.99 (t, *J* = 5.1 *Hz*, 1H, NH), 10.72 (s, 1H, NH); Mass (*m/z*): 431 (M^+^), 266, 251, 206, 182, 179, 161, 146, 133, 120, 106, 91 (100%, base peak), 78, 65 and 63.

##### 4-(2-(Benzo[d]oxazol-2-ylthio)acetamido)-N-phenethylbenzamide (**8l**)

IR: 3304 (NH), 3032 (C-H aromatic), 2922 (C-H aliphatic), 1633 (CO amide); ^1^H NMR: 2.85 (t, *J* = 7.2 *Hz*, 2H, CH_2_), 3.47 (m, 2H, CH_2_), 4.43 (s, 2H, CH_2_), 7.25 (m, 7H, Ar-H), 7.66 (m, 4H, Ar-H), 7.82 (d, *J* = 8.5 *Hz*, 2H, Ar-H), 8.50 (t, *J* = 5.3 *Hz*, 1H, NH), 10.69 (s, 1H, NH); Mass (*m/z*): 431 (M^+^), 327, 311, 192, 161, 146, 132, 120 (100%, base peak), 120, 104, 91, 77 and 65.

##### 4-(2-((5-Methylbenzo[d]oxazol-2-yl)thio)acetamido)-N-phenethylbenzamide (**8m**)

IR (KBr, cm^−1^): 3298 (NH), 3028 (C-H aromatic), 2924, 2865 (C-H aliphatic), 1663, 1631 (CO amide), 1535, 1350 (NO_2_), 1234 (C-O); ^1^H NMR: 2.40 (s, 3H, CH_3_), 2.84 (t, *J* = 7.2 *Hz*, 2H, CH_2_), 3.49 (m, 2H, CH_2_), 4.42 (s, 2H, CH_2_), 7.12 (d, *J* = 8.2 *Hz*, 1H, Ar-H), 7.20 (m, 3H, Ar-H), 7.28 (d, *J* = 7.2 *Hz*, 2H, Ar-H), 7.42 (s, 1H, Ar-H), 7.51 (d, *J* = 8.2 *Hz*, 1H, Ar-H), 7.66 (d, *J* = 8.5 *Hz*, 2H, Ar-H), 7.82 (d, *J* = 8.5 *Hz*, 2H, Ar-H), 8.51 (t, *J* = 5.4 *Hz*, 1H, NH), 10.70 (s, 1H, NH); Mass (*m/z*): 445 (M^+^), 341, 325, 271, 251, 206, 175, 164, 146, 136, 120 (100%, base peak), 104, 91, 78 and 65.

### 4.2. Biological Testing

#### 4.2.1. In Vitro Antitumor Assay

This test was carried out on three different human cancer cell lines: MCF-7, HCT116, and HepG2 using the MTT method [66]. The Appendix A provides further detailed information about the study.

#### 4.2.2. In Vitro VEGFR-2 Kinase Inhibitory Assay

VEGFR-2 enzyme linked immunosorbent assay (ELISA) kits were used. The Appendix A provides further detailed information about the study.

#### 4.2.3. HepG2 Cell Cycle Analysis

HepG2 Cell Cycle Analysis was performed for compound **8d** according to the methodology reported by Wang et al. [67]. The Appendix A provides further detailed information about the study.

#### 4.2.4. The effect of 8d on apoptosis and necrosis rates of HepG2

The effect of **8d** on apoptosis and necrosis rates of HepG2 was calculated using the Annexin V assay, according to the published technique [68,69]. The Appendix A provides further detailed information about the study.

#### 4.2.5. The effects of 8d on TNF-α and IL-6

The effects of **8d** on TNF-α and IL-6 were determined using the qRT-PCR technique [70,71]. The Appendix A provides further detailed information about the study.

#### 4.2.6. Molecular Docking

Molecular docking as carried out against VEGFR-2 (ID: 4ASD) using the Molecular Operating Environment (MOE) 2014 software [72,73,74,75,76]. The Appendix A provides further detailed information about the study.

## Data Availability

Data are available with corresponding authors upon request.

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
