# Peer review of "Modified Benzoxazole-Based VEGFR-2 Inhibitors and Apoptosis Inducers: Design, Synthesis, and Anti-Proliferative Evaluation"

_molecules, 2022, doi:10.3390/molecules27155047_

Round 1

Reviewer 1 Report

The manuscript entitled as “Modified Benzoxazole Based VEGFR-2 Inhibitors and 2 Apoptosis Inducers: Design, Synthesis, and Anti-Proliferative 3 Evaluationis an interesting manuscript that will be of interest to the readers of this journal which can definitely would encourage the scientific community working in this area of research. I would recommend this article for publication in the journal “Molecules”. However, the manuscript has some issues given below, that will need addressing before publication:

1        Explain the complications of current drug therapy with VEGFR-2 kinase inhibitors (including Sorafenib) in the introduction section.

2        Use scientific terms instead of layman (page no. 4, line 98, the word ‘boiling’ should be replaced by ‘reflux’. The article also has some typographical error and some grammatical errors. The authors should carefully check the manuscript for these errors. Some examples are given below:

Page no 4, line 105, ‘amid’ should be ‘amide’

Page no 4, line 109, ‘water path’ should be ‘water bath’

Page no. 5, Scheme 2, word ‘1,2 Dichloethane’ should be ‘1,2-dichloroethane’.

Page no 9, line 196, ‘compound 8 effect these cytokines…’ should be ‘compound 8 affect these cytokines…’

Replace the word ‘plain benzoxazole’ with ‘Unsubstituted benzoxazole’

Conclusion section, Page no. 14, line 261-263, ‘Moreover, compound 8d exhibited a significant…..compared to the dexamethasone (93.15%)’ need attention and should be rephrased.

3        Section 2.2.1., heading should be changed to ‘In vitro anticancer activity’ instead of ‘In vitro antitumor assay’.

4        Add standard deviation values to the data provided into Table 1 and Table 2. For the most potent compound (8d), the testing concentrations and their respective %age inhibition values against all the cancer cell lines and VEGFR-2, from which authors have calculated the IC50 values should be provided into the supplementary information.

5        Probable reason, why the electron withdrawing group (-NO2) showed better results than electron donating group (-CH3) at the terminal hydrophobic moiety should be provided in the SAR.

6        It would be better to correlate the SAR data to the results of molecular docking studies of potent compounds. Additionally, compound 8d showed better results than Sorafenib (standard drug). The authors should highlight the interactions that make the compound better than sorafenib.

After addressing these comments in the revised manuscript, the article can be published in the journal Molecules.

Author Response

Reviewer 1

The manuscript entitled as “Modified Benzoxazole Based VEGFR-2 Inhibitors and 2 Apoptosis Inducers: Design, Synthesis, and Anti-Proliferative 3 Evaluation” is an interesting manuscript that will be of interest to the readers of this journal which can definitely would encourage the scientific community working in this area of research. I would recommend this article for publication in the journal “Molecules”. However, the manuscript has some issues given below, that will need addressing before publication:

Response: Thank you for your efforts. All comments were considered in high interest and all changes were highlighted in the revised manuscript.

  • Explain the complications of current drug therapy with VEGFR-2 kinase inhibitors (including Sorafenib) in the introduction section.

Response: Done

  • Use scientific terms instead of layman (page no. 4, line 98, the word ‘boiling’ should be replaced by ‘reflux’. The article also has some typographical error and some grammatical errors. The authors should carefully check the manuscript for these errors. Some examples are given below:

Page no 4, line 105, ‘amid’ should be ‘amide’

Page no 4, line 109, ‘water path’ should be ‘water bath’

Page no. 5, Scheme 2, word ‘1,2 Dichloethane’ should be ‘1,2-dichloroethane’.

Page no 9, line 196, ‘compound 8 effect these cytokines…’ should be ‘compound 8 affect these cytokines…’

Replace the word ‘plain benzoxazole’ with ‘Unsubstituted benzoxazole’

Conclusion section, Page no. 14, line 261-263, ‘Moreover, compound 8d exhibited a significant…..compared to the dexamethasone (93.15%)’ need attention and should be rephrased.

Response: Done

  • Section 2.2.1., heading should be changed to ‘In vitro anticancer activity’ instead of ‘In vitro antitumor assay’.

Response: Done

  • Add standard deviation values to the data provided into Table 1 and Table 2. For the most potent compound (8d), the testing concentrations and their respective %age inhibition values against all the cancer cell lines and VEGFR-2, from which authors have calculated the IC50 values should be provided into the supplementary information.

Response: This point was clarified in supplementary data as per requested.

  • Probable reason, why the electron withdrawing group (-NO2) showed better results than electron donating group (-CH3) at the terminal hydrophobic moiety should be provided in the SAR.

Response: The electron withdrawing groups as NO2 and COCH3 are more beneficial for activity than the electron donating one as CH3. This may be due to decreased partial negative charge on the nitrogen of the amide moiety. This may increase the acidity of the NH group and consequently lead to increase its ability to form a hydrogen bond donor with the carbonyl moiety of Asp1046. In other words, the nitro group can do the same electronic effect that may be done by the two electron withdrawing groups (chloro and trifluoromethyl) of sorafenib. So that, compound 8d (incorporating a nitro group at para position of the terminal phenyl ring) has comparable activity with sorafenib.

  • It would be better to correlate the SAR data to the results of molecular docking studies of potent compounds. Additionally, compound 8d showed better results than Sorafenib (standard drug). The authors should highlight the interactions that make the compound better than sorafenib. After addressing these comments in the revised manuscript, the article can be published in the journal Molecules.

Response: additional explanation was added in the docking section to correlate the activity of compound 8d with its binding mode against VEGFR-2.

Reviewer 2 Report

Dear Author,

The manuscript title "Modified Benzoxazole Based VEGFR-2 Inhibitors and 2 Apoptosis Inducers: Design, Synthesis, and Anti-Proliferative 3 Evaluation" submitted by Alaa Elwan Et al., has provided some critical outcomes by derivatizing the known standard inhibitor to understand its pharmacophore.

Please address these minor points in your manuscript.

Fig 3: Y-axis annotation is missing

Fig 4: Y-axis annotation is missing

Molecular Docking- Should be performed along with the standard drug.

Docking score is not provided for each compound.

Molecular Dynamics studies could be added advantage.

SAR is missing.

All the best.

Author Response

Reviewer 2

The manuscript title "Modified Benzoxazole Based VEGFR-2 Inhibitors and 2 Apoptosis Inducers: Design, Synthesis, and Anti-Proliferative 3 Evaluation" submitted by Alaa Elwan Et al., has provided some critical outcomes by derivatizing the known standard inhibitor to understand its pharmacophore.

Thank you for your efforts and your valuable comments. All comments were considered in high interest and all changes were highlighted in the revised manuscript.

Please address these minor points in your manuscript.

  • Fig 3: Y-axis annotation is missing

Response: Added

  • Fig 4: Y-axis annotation is missing

Response: Added

  • Molecular Docking- Should be performed along with the standard drug.

Response: Docking study was carried out for sorafenib as reference drug.

  • Docking score is not provided for each compound.

Response: Done

  • Molecular Dynamics studies could be added advantage.

Response: thank you for this recommendation. It will take a long time not suitable for the available time of the revision process. If you persist, we can do it with an extension of time.

  • SAR is missing.

Response: SAR study was clarified in the revised manuscript.

Reviewer 3 Report

The manuscript - molecules-1851022-peer-review-v1.pdf - “Modified Benzoxazole Based VEGFR-2 Inhibitors and Apoptosis Inducers: Design, Synthesis, and Anti-Proliferative Evaluation” is chemically interesting and the article topic meets the scope of the journal. In this manuscript, thirteen new compounds based on benzoxazole moiety were synthesized and evaluated against three human cancer cell lines namely breast cancer (MCF-7), colorectal carcinoma 19 (HCT116), and hepatocellular carcinoma (HepG2). The synthesized compounds as well were evaluated against VEGFR-2 kinase activity by combining experimental and theoretical studies.

The authors present extensive experience in the field of the VEGFR-2 kinase study and anticancer agents, supported by published /cited articles (~ 25% self-citations) in well-known journals with a high impact factor. This manuscript provides well-planned scientific research leading to the discovery of bioactive derivatives based on benzoxazole moiety. The strategy applied to achieve their goals are adequate, very well structured, professionally interpreted, and very actual. Consequently, their work has promoted a high-quality scientific level.

 Please find enclosed four minor comments addressed to the manuscript authors:

 Comment 1. Please pay attention when writing the main text, e.g.:

- line 37, remove the space before [1] (please check and add or remove space from the entire manuscript);

- line 59, add a full stop after [42];

- line 138, replace capitalise the word “in” (please check all main text for such mistakes);

- lines 401-402, “The molecular docking study was carried out against VEGFR-2 (ID: 4ASD) using the Molecular Operating Environment (MOE) 2014 software” contradict the Supplementary Material, “The protein-ligand docking studies were carried out using MOE version 2019”,  etc.

Comment 2. The “Introduction” section of the manuscript requires extensive revision.  I would suggest that the authors attempt to present the key objectives of their study with regards to what is currently known (i.e. literature), thus highlighting the added value of the paper. The introduction is very small. Please increase its size in terms that the background story is underlined and adds impact to your work.

Comment 3. For Figure 6, I recommend providing in addition to the 2D representation and 3D representation of sorafenib interactions within the 4ASD binding site. Are only hydrogen bonds between the ligand and protein observed? Please identify and portray the hydrophobic (or other) interactions. Please provide a better quality of all Figures.

Comment 4. Relevant information is presented in the Supplementary Material. Please add in the main text the significant information which allows readers to easily find the information they are looking for.

This study is relevant and I consider it will be of interest to the scientific community, and in my opinion, after the mentioned minor issues are solved, the manuscript can be accepted for publication in the Molecules journal. 

Author Response

Reviewer 3

The manuscript - molecules-1851022-peer-review-v1.pdf - “Modified Benzoxazole Based VEGFR-2 Inhibitors and Apoptosis Inducers: Design, Synthesis, and Anti-Proliferative Evaluation” is chemically interesting and the article topic meets the scope of the journal. In this manuscript, thirteen new compounds based on benzoxazole moiety were synthesized and evaluated against three human cancer cell lines namely breast cancer (MCF-7), colorectal carcinoma 19 (HCT116), and hepatocellular carcinoma (HepG2). The synthesized compounds as well were evaluated against VEGFR-2 kinase activity by combining experimental and theoretical studies.

The authors present extensive experience in the field of the VEGFR-2 kinase study and anticancer agents, supported by published /cited articles (~ 25% self-citations) in well-known journals with a high impact factor. This manuscript provides well-planned scientific research leading to the discovery of bioactive derivatives based on benzoxazole moiety. The strategy applied to achieve their goals are adequate, very well structured, professionally interpreted, and very actual. Consequently, their work has promoted a high-quality scientific level.

 Please find enclosed four minor comments addressed to the manuscript authors:

Thank you for your praise and valuable comments. I considered these comments with high interest. The comments and my response are summarized in the following points.

  • Please pay attention when writing the main text, e.g.: line 37, remove the space before [1] (please check and add or remove space from the entire manuscript);

Response: Done

  • line 59, add a full stop after [42];

Response: Done

  • line 138, replace capitalise the word “in” (please check all main text for such mistakes);

Response: Done

  • lines 401-402, “The molecular docking study was carried out against VEGFR-2 (ID: 4ASD) using the Molecular Operating Environment (MOE) 2014 software” contradict the Supplementary Material, “The protein-ligand docking studies were carried out using MOE version 2019”,  etc.

Response: we apologize for this typo. It is corrected in supplementary data.

  • The “Introduction” section of the manuscript requires extensive revision.  I would suggest that the authors attempt to present the key objectives of their study with regards to what is currently known (i.e. literature), thus highlighting the added value of the paper. The introduction is very small. Please increase its size in terms that the background story is underlined and adds impact to your work.

Response:

  • For Figure 6, I recommend providing in addition to the 2D representation and 3D representation of sorafenib interactions within the 4ASD binding site. Are only hydrogen bonds between the ligand and protein observed? Please identify and portray the hydrophobic (or other) interactions. Please provide a better quality of all Figures.

Response: 2D and 3D interaction of sorafenib was added. the different bonds were clarified. The figures were improved as per requested. 

  • Relevant information is presented in the Supplementary Material. Please add in the main text the significant information which allows readers to easily find the information they are looking for.

Response: Thank you for this recommendation. However, our institution has a big restriction for the percent of plagiarism in each published article (not more than 25%). If we added the different methods in the main manuscript, the percent would increase more than 25% and this will make a certain problem in out institution. Hope you understand our situation.

This study is relevant, and I consider it will be of interest to the scientific community, and in my opinion, after the mentioned minor issues are solved, the manuscript can be accepted for publication in the Molecules journal.